# What Drives Visitors’ Use of Bins in Urban Parks? An Application of the Stimulus-Organism-Response Model

**DOI:** 10.3390/ijerph192114170

**Published:** 2022-10-29

**Authors:** Pengwei Wang, Lirong Han, Fengwei Ai

**Affiliations:** 1School of Tourism, Shanghai Normal University, Shanghai 200234, China; 2School of Geography and Tourism, Hulunbeier University, Hulunbuir 021008, China

**Keywords:** facilitators, inhibitors, littering behaviour, personal norms, social norms

## Abstract

Littering by visitors has led to severe challenges for rubbish collection in urban parks. One way to solve this problem is to encourage visitors to put rubbish in the bin. The purpose of this study is to explore the mechanism that drives people’s use of bins in urban parks. The theoretical model of stimulus-organism-response is used to test the influence of stimuli (personal and social norms) on people’s psychology (facilitators and inhibitors), thereby producing responses (the use of bins). In this study, we used a purposeful sampling method. Overall, 400 questionnaires were distributed, and 356 valid questionnaires were collected from visitors to the Shanghai City Park in China. The data were analysed using structural equations. The results show that personal and social norms have a significant impact on visitors’ internal psychological state (facilitators and inhibitors). More specifically, personal and social norms are positively correlated with facilitators and negatively correlated with inhibitors. They have a significant positive impact on people’s use of bins. We also found that facilitators and inhibitors partially mediate the relationship between norms and behaviours. The study suggests park managers should introduce various measures to influence people’s personal norms and cultivate people’s awareness of their obligation, responsibility, and commitment to the environment, and managers should also show visitors the consequences of not properly disposing of their rubbish as well as place more rubbish bins in key areas.

## 1. Introduction

Pro-environmental behaviour is behaviour that does not waste natural resources or improves the quality of the environment [1,2,3]. It has been widely studied in relation to the tourism industry, with research focusing on green restaurants [4], museums [5], cruises [6], religious tourism [7], hotels [8,9,10], ecotourism destination [11], national parks [12], zoos/aquariums [13], and more. Pro-environmental behaviour refers to different things in different research contexts. Previous studies have considered what shapes pro-environmental behaviour, but the fact that pro-environmental behaviour has been studied as a universal activity has led to a limited understanding of this complex behaviour. To make research into pro-environmental behaviour more reliable, Goh et al. (2017) suggested focusing on a single pro-environmental behaviour in a specific context [14]. Therefore, in this study, we look solely at the use of bins, treating this as one aspect of pro-environmental behaviour. Visitors who use bins are those who put their own rubbish into a bin (if provided) or put it into a bag or pocket before putting it into a bin later.

Previous studies have examined the process by which visitors develop the intention to use bins. These studies have focused on tourist destinations such as national parks, nature reserves, riversides, beaches, and mountains [15,16,17]. However, there have been relatively few studies of the factors that affect the use of bins in urban parks.

Urban parks are different from national parks, nature reserves, and other tourist destinations. They have unique features and functions, and they play an important role in the life of urban citizens [18]. Especially since the start of the COVID-19 pandemic, travel between regions has been limited. Thus, urban parks have become the main leisure destinations for many city dwellers [19,20,21]. An increase in the number of park visitors has led to significant challenges regarding the collection and disposal of rubbish. Littering not only detracts from the park’s visual quality but can also be harmful to animals, the soil, water, and other people. Therefore, it is necessary to understand why visitors do or do not put their rubbish in the bin. This would have a significant impact on the collection and treatment of rubbish in urban parks. It would also help to reduce labour costs in parks and preserve their natural beauty.

Previous studies have used theories from social psychology to explain the pro-environmental behaviour of individuals. These theories include the norm activation model [22], theory of reasoned action [23], theory of planned behaviour [24], value-belief-norm theory [25], and the goal-oriented behaviour model [26]. In recent years, some studies have integrated two or more social psychology theories to explain individuals’ pro-environmental behaviours [27,28]. For example, Esfandiar et al. (2021) combined the planned behaviour theory and the norm activation model to study visitors’ use of bins in Yanchep National Park [29].

We used the theory of stimulus-organism-response (SOR) [30] to explain visitors’ use of bins. The SOR model originated in the field of psychology. It is mainly used to explain the influence of environmental factors on the psychological and behavioural responses of individuals. The model consists of three parts: stimulus, organism, and response. The stimulus (S) is comprised of the variables that exist in the environment surrounding an individual. The organism (O) refers to the cognitive evaluations and psychological processes of an individual or organism. The response (R) is the decision that is taken based on the interaction between the stimulus and the organism. This model provides a mechanism for explaining complex decisions. It clarifies complex human behaviours [31]. Therefore, the SOR model has been widely used in explaining pro-environmental behaviours and wasteful behaviours [32,33,34,35]. They have been used to explain why people purchase green products [36] or natural products [37]. However, no studies have used this model to explain the use of bins.

This study attempts to answer the following research questions. Q1: What are the antecedents of park visitors’ positive and negative evaluation of the use of bins? Q2: How do park visitors’ positive or negative assessments of the use of bins promote or inhibit their use of them? Q3: How do facilitators and inhibitors mediate the relationship between antecedents and visitors’ use of bins?

This study is innovative in the following ways. First, it uses the SOR framework to investigate urban park visitors’ decision making regarding the use of bins. In this way, it develops the literature regarding city park management and SOR. It demonstrates that SOR can be a productive model for explaining complicated decision-making processes and has been effectively used to explain other environmental behaviours [36,37,38] but it is underused in explaining visitors’ use of bins. Second, previous studies aimed at understanding the direct effects of personal norms and social norms on use of bins, and this study also explores the mediation role of facilitators and inhibitors between norms and behaviours. The results of this study will help park managers in China and other parts of the world to understand how to encourage urban park visitors to use bins more regularly and more effectively.

The research model is shown in Figure 1. In terms of our SOR model, the stimuli were personal and social norms; the internal states of organisms were the facilitators and inhibitors of the use of bins; and the use of bins was the potential response.

## 2. Development of Hypotheses

### 2.1. Personal Norms, Facilitators, Inhibitors, and the Use of Bins

Personal norms reflect people’s tendency to behave a certain way in specific situations. Those tendencies correlate with what they perceive to be good or bad [39]. If people think that a certain behaviour is consistent with their personal norms, they will engage in that behaviour [40]. The role played by personal norms in influencing decisions has been widely studied in relation to environmentally friendly behaviours. For example, focusing on tourism, Wu et al. (2022) demonstrated the importance of personal norms in prompting visitors to make environmentally responsible decisions [41]. Wang et al. (2021) proved that personal norms play a key role in encouraging visitors to reduce waste [42]. Esfandiar et al. (2021) revealed the important role of personal norms in visitors’ use of bins in national parks [29]. Based on this, we put forward the following hypotheses:

**H1.** 
*Visitors’ personal norms are positively correlated with the factors that facilitate the use of bins.*


**H2.** 
*Visitors’ personal norms are negatively correlated with the factors that inhibit the use of bins.*


**H3.** 
*Visitors’ personal norms are positively correlated with their use of bins.*


### 2.2. Social Norms, Facilitators, Inhibitors, and Use of Bins

Social norms are the common beliefs and behavioural standards that groups adopt. They are not legally enforced, but they have an influence on people’s actions [43]. Individuals want to be recognized or liked by other people, and social norms play an important role in shaping their behaviours as they navigate these desires [32]. Previous studies have shown that social norms can influence individuals’ decision making [44], especially when it comes to environmental choices. For example, Zhang et al. (2018) proved that social norms play a role in people’s intention to use a car when visiting a national park [45]. Meanwhile, Ojedokun et al. (2022) found that social norms are a predictor of the intention to prevent littering and actual littering-prevention behaviour [46]. Therefore, we propose the following hypotheses:

**H4.** 
*Visitors’ social norms are positively correlated with the factors that facilitate visitors’ use of bins.*


**H5.** 
*Visitors’ social norms are negatively correlated with the factors that inhibit visitors’ use of bins.*


**H6.** 
*Visitors’ social norms are positively correlated with their use of bins.*


### 2.3. Facilitators, Inhibitors, and the Use of Bins

Our research shows that visitors are able to identify positive and negative reasons for using a bin. The positive reasons encourage them to use bins, whereas the negative reasons discourage them from using bins. According to behavioural reasoning theory (BRT), individuals will make a decision after weighing the reasons for and against it [47,48]. We believe that the facilitators of putting rubbish into the bin (such as a concern for protecting the environment of a park, along with its animals and plants) will have a positive impact on visitors’ use of bins. By contrast, the inhibitors of putting rubbish into a bin (such as the inconvenience of picking up the rubbish, carrying it, and finding a bin) may limit visitors’ use of bins. BRT has been proven in research concerning consumer decision making [47,48,49]. Therefore, we assume that:

**H7.** 
*The facilitators of putting rubbish into a bin are positively related to visitors’ use of bins.*


**H8.** 
*The inhibitors of putting rubbish into a bin are negatively correlated with visitors’ use of bins.*


### 2.4. Mediation Effect of Facilitators and Inhibitors

This research provides a deeper understanding of visitors’ use of bins by examining the indirect relationship between norms and behaviours as well as the direct relationship. Clarifying the relationship between norms and behaviours will make it easier to explain the complex decision making relating to visitors’ use of bins. It will explain the mediating effects of facilitators and inhibitors on norms and behaviours. We propose the following hypotheses:

**H9.** 
*The facilitators of visitors’ use of bins will mediate the relationship between their use of bins and (a) personal norms and (b) social norms.*


**H10.** 
*The inhibitors of visitors’ use of bins will mediate the relationship between their use of bins and (a) personal norms and (b) social norms.*


## 3. Study Methods

### 3.1. Study Site

Shanghai is the largest city in China in terms of its economy. It is China’s centre for innovation regarding international affairs, finance, trade, shipping, and technology. It has a total area of 6340.50 km^2^, spread over 16 districts. At the end of 2021, the resident population of the city was 24,894,300, and the regional GDP was RMB 4,321,485 million. In 2018, the “Shanghai Urban Master Plan” set out the goal of transforming Shanghai into an ecological city and making it a “city with 1000 gardens.”

At the end of 2021, there were 532 urban parks in Shanghai. On the seven-day National Day holiday of the Republic of China in 2021, 4,984,100 people visited the city’s urban parks. Shanghai’s urban parks have become its residents’ destinations of choice when it comes to leisure and recreation. We selected three urban parks as our research sites: Shanghai Century Park, Daning Lingshi Park, and Changfeng park. All of them are larger than 20 ha. According to the classification of urban parks created by Wood et al. (2017) [50], parks that are larger than 20 ha are regional parks. These have more visitors that come from further afield. Therefore, they were more suitable for us as case sites. As well as size, we also chose the parks based on the following selection criteria: each park was located near the centre of the city; each park belonged to a different district; each park was well-established; each park was very popular; and each park received a large number of visitors.

### 3.2. Qualitative Interviews

We conducted qualitative interviews to understand every aspect of why visitors decide to put their rubbish in the bin. Firstly, we reviewed the literature to generate the interview guide. Then, two professors of environmental behaviour and one park manager were consulted to improve the interview guide further.

After the guide was prepared, we interviewed 20 park visitors face to face in September 2021, including 9 visitors from Shanghai Century Park, 5 visitors from Daning Lingshi Park, and 6 visitors from Changfeng park. The interviewees included 11 males and 9 females, aged between 20 and 66, who were frequent visitors to the park.

The questions were as follows: (a) What made you realize that you should put your rubbish into the bin? (b) When playing in the park, how do your friends and the people around you deal with their rubbish? Does the way they deal with rubbish affect you? (c) What impact do you think littering has when you are playing in the park? (d) What makes you decide not to put rubbish in the bin sometimes? (e) Generally, what do you think of the use of bins while you are playing in the park?

First, we identified the stimulus. The results of the interviews showed that personal and social norms both shape visitors’ views on the use of bins. Most people stated that it was everybody’s responsibility to protect the park and put their rubbish in the bin. They also stated that it was a standard behaviour in society. Based on these findings, we proposed personal and social norms as stimuli, an idea that was consistent with previous literatures. The literature has noticed that personal norms and social norms influence visitors’ pro-environmental behaviour [45,51] and use of bins [29].

Second, we identified the relevant organisms. These were the factors that facilitated or inhibited the use of bins. The aim was to capture the psychological processes of park visitors when they were deciding whether to put their rubbish in the bin. In the interviews, people tended to describe the negative effects of littering and the fact that it was inconvenient to put rubbish in the bin. Therefore, we took the negative consequences of not putting rubbish in the bin as facilitators; these were factors that convinced people to put rubbish in the bin. On the other hand, we took the benefits of not putting rubbish in the bin as inhibitors; these were the factors that made people reluctant to put rubbish in the bin. These facilitators and inhibitors act as a link between stimuli and responses. This demonstrates that visitors’ norms have a direct effect on their use of bins, but visitors also use reasoning (through facilitators and inhibitors) to decide what to do before acting.

Finally, we took visitors’ use of bins as their responses. These were the responses based on their negative or positive assessment of putting their rubbish in the bin. We used “on-site behaviour” instead of intention as the result variable. Although the response of visitors’ on-site use of bins differs from their “actual use of bins”, it was possible to collect more reliable data by asking people about their current use of bins rather than their future intentions.

As well as direct impact, we also measured the impact of the interaction of related variables on visitors’ decision making. We studied the mediating role played by facilitators and inhibitors in the link between people’s norms and their use of bins.

### 3.3. Measurement Instrument

We combined our qualitative research results with scales used in previous studies to measure every dimension. Personal norms were measured using a five-item scale that was adapted from Esfandiar et al. (2021) [29]. Social norms were measured using a three-item scale that was adapted from Han et al. (2017) [52]. An adapted three-item scale was used to measure the use of bins [15]. Three scales developed by our qualitative research were used to measure the facilitators and inhibitors for the use of bins. These are shown in Table 1. We used a seven-point Likert scale, with answers ranging from 1 (strong disagreement) to 7 (strong agreement).

### 3.4. Data Collection and Analysis Approach

After we developed the initial questionnaire, we tested its validity. First, we asked for advice from three experts, including two professors who studied pro-environmental behaviours and a practitioner who worked in a city park. Based on their suggestions, we revised some items in the questionnaire. We then conducted a preliminary test with 20 visitors to Century Park to check that the questionnaire was readable.

The final questionnaire was made up of three parts. In the first part, a seven-point Likert scale was used to measure visitors’ personal norms, social norms, facilitators, and inhibitors. The second part contained used a seven-point Likert scale to measure visitors’ use of bins. The third part gathered data about the demographics of the visitors, including gender, age, income, and education.

The questionnaire was distributed face-to-face by three trained investigators in Shanghai Century Park, Daning Lingshi Park, and Changfeng park. The survey was conducted in October 2021. To increase the representativeness of the respondents and overcome the problems with random sampling in urban parks, we used a purposeful sampling method that took the following factors into account: gender, age, types of visiting groups (individuals, groups, and families), and types of recreational activities. In total, 400 questionnaires were distributed, and 356 valid questionnaires were collected. According to Kock and Hadaya (2018) and Esfandiar et al. (2021), the sample size should be greater than 10 times the maximum number of inner or outer model paths in the structural model [15,53]. The number of paths in the structural model is eight (Figure 1), which shows our sample size is sufficient for the SEM analysis. Table 2 shows the demographics of the respondents.

We used structural equation modelling (SEM) to analyse the data in SPSS 26 and AMOS 26. We followed the two-step procedure of generating the measurement model and evaluating the structural path. Before the SEM, we checked the abnormal value and normality of the data. The data were normally distributed because the skewness and kurtosis values were lower than the prescribed thresholds [54]. We also checked and confirmed that there were no problems with multicollinearity [55]. Finally, we performed mediation analysis using Process Macro.

## 4. Results

### 4.1. Demographic Profile of Respondents

The socio-economic characteristics of the respondents are shown in Table 2. The age distribution was fairly typical, and there were slightly more female visitors than male visitors. More than half of the respondents had completed or were pursuing a bachelor’s degree. The income of the respondents was fairly typical, with a small proportion of respondents earning more than RMB 15,000/month.

### 4.2. Common Method Bias (CMB)

Self-reported data were used in this study, so there may have been common method bias (CMB) [56]. To reduce the possibility of CMB, the names of the participants were anonymised. Different sources were used for items, and the order of the items was altered. In addition, we tested the CMB by using common factor analysis, and the fitting index of the model was poor (χ^2^/df = 23.96, CFI = 0.566, NFI = 0.557, IFI = 0.567, RFI = 0.493, TLI = 0.504, RMSEA = 0.254).

### 4.3. Measurement Model

We used confirmatory factor analysis (CFA) to test reliability and validity. The reliability of each construct (Cronbach’ s alpha) was greater than 0.7, which is in line with the recommended threshold [57]. We used factor loadings, their composite reliability (CR), and average variance extracted (AVE) to measure convergent validity. The factor loadings of the items were above 0.50 (Table 1), meeting the recommended standard [58]. The CR values of every construct were above 0.70, meeting the recommended threshold (Table 1) [58]. The AVE also reached the recommended threshold of 0.50 (Hair et al., 1998; Table 1) [59]. The square root of the AVE of all the constructs exceeded the correlation between the pair of structures, confirming the discriminant validity (Table 3) [60]. We looked at the values of the CFI, NFI, IFI, RFI, TLI, and RMSEA for the assessment of structural model fit. The result indicated that the hypothesised model returned a good model fit (χ^2^/df = 1.49, CFI = 0.99, NFI = 0.97, IFI = 0.99, RFI = 0.97,TLI = 0.99, RMSEA = 0.04).

### 4.4. Control Variables

We controlled for the effect that demographic factors (age, gender, education, and income) had on the use of bins. The results indicated that no socio-demographic factors had a significant influence on the use of bins.

### 4.5. Structural Model and Hypotheses Testing

The degree of fit of the structural model was good (χ^2^/df = 2.12, CFI = 0.99, NFI = 0.99, IFI = 0.99, RFI = 0.97, TLI = 0.99, RMSEA = 0.056). The results show that visitors’ personal norms were positively correlated with the facilitators of the use of bins (H1 and H3), and their personal norms were negatively correlated with the inhibitors of the use of bins (H2). The statistical results support H1 (β = 0.38, *p* < 0.001), H2 (β = −0.18, *p* < 0.05), and H3 (β = 0.37, *p* < 0.001). Similarly, social norms were positively correlated with the facilitators of the use of bins (H4 and H6) and negatively correlated with the inhibitors of the use of bins (H5). The analysis of the statistical results supported H4 (β = 0.42, *p* < 0.001), H5 (β = −0.22, *p* < 0.01), and H6 (β = 0.17, *p* < 0.001). In addition, the facilitators were positively correlated with the use of bins (H7), and the inhibitors were negatively correlated with the use of bins (H8). Both hypotheses are supported. The results show that H7 (β = 0.42, *p* < 0.001) and H8 (β = −0.05, *p* < 0.05) are supported. The research model showed 46.9% variance for the facilitators, 7.6% for the inhibitors, and 68.6% for the use of bins. The results are shown in Figure 2 and Table 4.

### 4.6. Mediation Analysis

We used model 4 in SPSS Process Macro to run parallel mediation analysis and test the mediating role played by facilitators and inhibitors in the relationship between norms and behaviours. The results show that facilitators and inhibitors partially mediated the relationship between people’s personal and social norms and their use of bins (Table 5 and Table 6).

## 5. Discussion and Implications

### 5.1. Discussion

This study has solved three research problems. In response to Q1, we evaluated the antecedents of the facilitators and inhibitors that stimulated visitors’ use of bins. We also tested the direct correlation between personal norms and social norms and the use of bins. Our results show that personal norms have a significant positive impact on facilitators (H1) and the use of bins (H3). This supports previous studies of environmental protections and eco-friendly behaviour [61]. This shows that people’s understanding of their obligation to put rubbish in the bin when playing in the park is positively correlated with the factors that facilitate their use of bins. This is driven by social and personal norms. In other words, visitors’ personal norms increase their awareness of the need to protect the environment. They understand that not putting rubbish in the bin will damage the environment, pose a threat to the park’s animals and plants, and increase the cost of maintaining the park.

Personal norms are negatively correlated with inhibitors (H2), which means that people’s sense of responsibility about putting rubbish in the bin will affect how their inhibitors affect their use of bins. This indicates that visitors’ personal norms can reduce their inhibitions about using bins, making them less concerned about the inconvenience caused by putting rubbish in a bin.

This study also shows that social norms have a statistically significant impact on visitors’ facilitators and inhibitors as well as their use of bins. Thus, H4, H5, and H6 are supported. This result is consistent with previous studies of the impact of social norms on environmental protections and eco-friendly behaviours [45,62,63]. These findings show that when visitors see that the people who are most important to them put their rubbish in the bin, it will prompt them to use bins themselves because they will internalize the relevant facilitators. They will consider the fact that not putting rubbish in the bin will damage the park, pose a threat to the park’s animals and plants, and increase the cost of maintaining the park. When they are influenced by people who are important them, people do not care as much about the inconvenience of putting their rubbish in a bin.

Personal norms and social norms have significant influence on people’s psychology and behaviour, which is consistent with the studies in other parts of the world [33,34]. Therefore, managers of relevant departments in countries all over the world should pay attention to the role of social norms and personal norms in improving people’s pro-environmental behaviour.

Q2 concerned the relationship between visitors’ evaluation of the positive and negative aspects of using bins and their use of bins. The results show that the facilitators of putting rubbish in a bin are positively correlated with using bins, whereas the inhibitors of putting rubbish in bin are negatively correlated with using bins. These findings confirm H7 and H8. The support for H7 means that visitors will put rubbish in a bin if they think that it will help protect the environment, protect the park’s animals and plants, and reduce the cost of maintaining the park. Similarly, people will be less likely to use bins if they find it inconvenient to pick up rubbish, carry rubbish, and look for bins.

In response to Q3, this study has found that the facilitators and inhibitors played a mediating role between norms and behaviours. More specifically, the results show that the facilitators and inhibitors of putting rubbish in bins partially mediate the relationship between personal norms and social norms and the use of bins. Thus, H9 (a), H9 (b), H10 (a), and H10 (b) are supported. These findings are consistent with our predictions and the results of previous environmental studies [64,65].

### 5.2. Theoretical Implications

This study has enriched the literature concerning environmental responsibility in two ways. First of all, although previous studies have used theory of planned behaviour and the norm activation model to explain the influence of personal and social norms on visitors’ use of bins, this study examined their use of bins by focusing on psychological processes. Thus, it has enriched the existing literature on the use of bins and provided new insights into visitors’ use of bins.

Secondly, we used the SOR as a theoretical model to examine how the facilitators and inhibitors of putting rubbish into bins were influenced by environmental stimuli. As far as we know, this study is the first attempt to use the SOR model to explain the use of bins by visitors. Thus, it enriches the literature concerning SOR, opening up a new context for future researchers.

### 5.3. Managerial Implications

This study has three useful implications for urban park managers and decision makers, which can provide reference for park management in China and other parts of the world. First, it shows that personal and social norms are the main driving forces when it comes to the use of bins. They have a significant impact on both the facilitators and inhibitors of people’s use of bins. Park managers should introduce various measures to influence people’s personal norms. They should cultivate people’s awareness of their obligation, responsibility, and commitment to the environment. Furthermore, they should encourage the idea that everyone is responsible for protecting the environment. The government should publicize the idea that it is every citizen’s responsibility to dispose of their rubbish. They should use online and offline channels such as social media and public service advertisements. They should establish a common social belief in the importance of protecting the environment and creating a civilized social space. At the same time, they should encourage people to provide an example to others. It is especially important for parents to set an example for their children.

Secondly, park managers should increase visitors’ understanding of their facilitators. They should show how rubbish is generated through explanations, videos, information panels, signs, and more. They should also show visitors the consequences of not properly disposing of their rubbish. They should emphasize the positive effects of adopting the right behaviours, thereby increasing people’s awareness about the consequences of littering in the park.

Finally, park managers should decrease the influence of inhibitors. They should place more rubbish bins in key areas, such as the entrances to parks, picnic areas, entertainment areas, etc. To prevent the landscape from being blemished by too many rubbish bins, the managers should provide machines to get rubbish bags for visitors when there are no bins nearby. This would mean that visitors do not have the inconvenience of having to carry their rubbish to a bin.

### 5.4. Limitations and Future Researches

As with all empirical studies, there are some limitations to the present research. First, this study relied on self-reported behaviour rather than observed behaviour for its data. Therefore, the data may have suffered from the bias of social expectation. Future studies could use observed behaviour to expand our research. Second, we only collected and analysed data from three urban parks in Shanghai, which limits the universality of our findings. Future studies could replicate our research and compare the results from other urban parks to test the robustness of our model. Finally, although this study explored several factors that might influence people’s use of bins, there were several factors that it did not consider. Future studies could examine other variables than the ones studied here, such as the influence of people’s attachment to a specific place and the environmental policies in the place, such as environmental warning signs, policies, rewards, punitive measures, and signs that encourage environmental behaviour.

## 6. Conclusions

This study used an SOR model to identify how the psychology of urban park visitors (based on facilitators and inhibitors) influenced their use of bins in response to certain stimuli (such as personal and social norms). Conclusions are drawn, and suggestions are made as follows:(1)The results show that personal and social norms have a significant impact on visitors’ facilitators and inhibitors and their use of bins. Visitors’ facilitators and inhibitors have significant impact on visitors’ behaviours. Facilitators and inhibitors also mediate norms and behaviours.(2)On the theoretical contribution, the study used the SOR as a theoretical model to examine how the facilitators and inhibitors of putting rubbish into bins were influenced by environmental stimuli. The study enriches the literature concerning SOR, opening up a new context for future researchers.(3)This study also made several proposals for urban park managers. Park managers should introduce various measures to influence people’s personal norms and cultivate people’s awareness of their obligation, responsibility, and commitment to the environment. They should also show visitors the consequences of not properly disposing of their rubbish and place more rubbish bins in key areas. The results of this study will help park managers in China and other parts of the world to understand how to encourage urban park visitors to use bins more regularly and more effectively.

## Figures and Tables

**Figure 1 ijerph-19-14170-f001:**
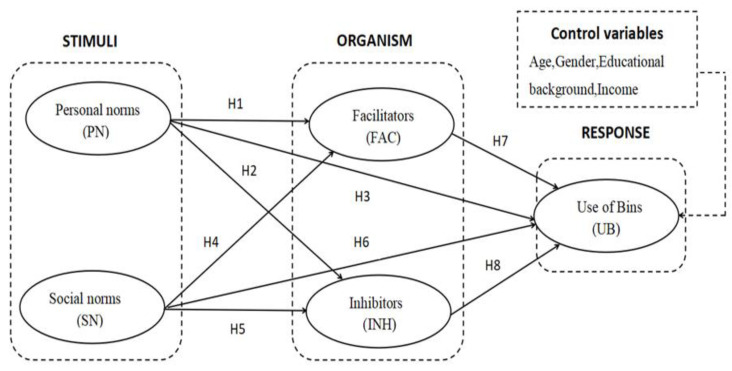
Proposed research model.

**Figure 2 ijerph-19-14170-f002:**
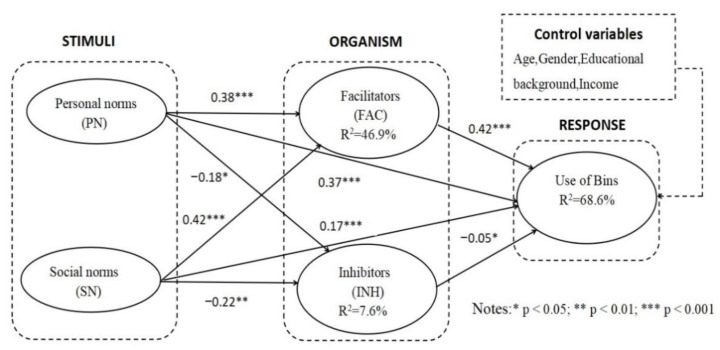
Result of hypotheses testing.

**Table 1 ijerph-19-14170-t001:** Results of internal reliability and validity.

Construct	Items	Internal Reliability	Convergent and Discriminant Validity		
		Cronbach’s Alpha (α)	Factor Loadings	CR	AVE
Personal norms (PN)	I feel a personal obligation to bin my litter while visiting the park.	0.950	0.906	0.951	0.795
	I feel guilty if I don’t bin my litter while visiting the park.		0.842		
	I feel morally obliged to minimise human impacts on parks by not littering.		0.894		
	Everybody should share the responsibility to bin their litter while visiting the park.		0.920		
	I feel responsible to clean up after a picnic in the park.		0.893		
Social norms (SN)	Most people who are important to me think I should place litter in the bin.	0.824	0.844	0.839	0.639
	Most people who are important to me would want me to place litter in the bin.		0.875		
	Most people who are important to me would prefer me to place litter in the bin.		0.662		
Facilitators (FAC)	Littering in the park destroys the park environment.	0.915	0.865	0.916	0.786
	Littering in the park poses a threat to the animals and plants.		0.914		
	Littering in the park increases the cost of park maintenance.		0.865		
Inhibitors (INH)	Not putting rubbish into the bin saves me the inconvenience of cleaning up litter.	0.976	0.959	0.976	0.932
	Not putting rubbish into the bin saves me the inconvenience of carrying litter.		0.967		
	Not putting rubbish into the bin saves me the inconvenience of looking for the bin.		0.970		
Use of bins (UB)	I place my empty cans, bottles, etc., in a bin while visiting the park.	0.940	0.948	0.942	0.843
	After a picnic in this park, I leave the place clean.		0.894		
	If there is no bin, I will take it away and put it into the bin later.		0.912		

Note: Composite reliability, CR; average variance extracted, AVE.

**Table 2 ijerph-19-14170-t002:** Demographic profile of respondents.

	Socio-Demographic Profile	Frequency	Percentage
Age	18−30 years	73	20.5%
	31−40 years	76	21.3%
	41−50 years	86	24.2%
	51−59 years	48	13.5%
	≥60 years	73	20.5%
Gender	Male	159	44.7%
	Female	197	55.3%
Educational background	Completed high school or below.	21	5.9%
	Completed/pursuing professional/vocational school	100	28.1%
	Completed/pursuing bachelor’s	191	53.7%
	Completed/pursuing master’s or higher	44	12.4%
Income (RMB/month)	<3000	61	17.1%
	3000−6000	83	23.3%
	6001−8000	81	22.8%
	8001−10,000	50	14%
	10,001−15,000	59	16.6%
	>15,000	22	6.2%

**Table 3 ijerph-19-14170-t003:** Validity and reliability analysis.

	Mean	SD	PN	SN	FAC	INH	UB
PN	6.14	0.96	**0.89**				
SN	6.01	1.04	0.57	**0.80**			
FAC	5.78	1.03	0.59	0.62	**0.89**		
INH	5.81	1.30	−0.23	−0.25	−0.24	**0.97**	
UB	6.16	1.03	0.71	0.65	0.75	−0.29	**0.92**

Note: Standard deviation, SD; use of bins, UB; personal norms, PN; social norms, SN; facilitators, FAC; inhibitors, INH. The values mentioned in bold represent the square roots of AVEs.

**Table 4 ijerph-19-14170-t004:** Hypotheses testing.

Hypothesis	Path	Estimate	*p*	Support
H1	PN−FAC	0.38	<0.001	Yes
H2	PN−INH	−0.18	<0.05	Yes
H3	PN−UB	0.37	<0.001	Yes
H4	SN−FAC	0.42	<0.001	Yes
H5	SN−INH	−0.22	<0.01	Yes
H6	SN−UB	0.17	<0.001	Yes
H7	FAC−UB	0.42	<0.001	Yes
H8	INH−UB	−0.05	<0.05	Yes

Note: Use of bins, UB; personal norms, PN; social norms, SN; facilitators, FAC; inhibitors, INH.

**Table 5 ijerph-19-14170-t005:** Results of mediation analysis.

PN → FAC/INH → UB						
	β	se	t	p	LLCI	ULCI
PN → FAC	0.62	0.05	13.07	0.000	0.5269	0.7135
PN → INH	−0.32	0.07	−4.37	0.000	−0.4617	−0.1751
PN → UB	0.44	0.04	10.45	0.000	0.3546	0.5191
FAC → UB	0.50	0.04	12.89	0.000	0.4238	0.5763
INH → UB	−0.06	0.03	−2.44	0.015	−0.1114	−0.0121
Total effect of PN → UB	0.77	0.04	18.23	0.000	0.6839	0.8493
**S** **N → FAC/INH → U** **B**						
	**β**	**se**	**t**	**p**	**LLCI**	**ULCI**
SN → FAC	0.60	0.04	14.31	0.000	0.5145	0.6784
SN → INH	−0.32	0.07	−4.94	0.000	−0.4529	−0.1950
SN → UB	0.29	0.04	6.79	0.000	0.2041	0.3704
FAC →UB	0.55	0.04	12.86	0.000	0.4678	0.6368
INH → UB	−0.07	0.03	−2.38	0.018	−0.1188	−0.0114
Total effect of SN → UB	0.64	0.04	15.60	0.000	0.5573	0.7181

Note: Use of bins, UB; personal norms, PN; social norms, SN; facilitators, FAC; inhibitors, INH.

**Table 6 ijerph-19-14170-t006:** Indirect effects between dependent and independent variable.

	β	se	LLCI	ULCI
PN → FAC → UB	0.31	0.03	0.2426	0.3786
PN → INH → UB	0.02	0.01	0.0044	0.0362
SN → FAC → UB	0.33	0.04	0.2551	0.4077
SN → INH → UB	0.02	0.01	0.0050	0.0380

Note: Use of bins, UB; personal norms, PN; social norms, SN; facilitators, FAC; inhibitors, INH.

## Data Availability

The data presented in this study are available on request from the corresponding author.

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
