# Peer review of "What Drives Visitors’ Use of Bins in Urban Parks? An Application of the Stimulus-Organism-Response Model"

_ijerph, 2022, doi:10.3390/ijerph192114170_

Round 1

Reviewer 1 Report

The paper needs a content reorganization. For example, point 2.2. is more similar to a methodological section rather that a literature review one.

As far as the section 2.3 is concerned, I think that much of the information is more proper of the theoretical section. Thus, I find some confusion in the content presentation. I also think that section 4 is divides into too many sections, and that the discourse should be more fluid.

I would advise authors to review the content and to reinforce the theoretical review.

Conclusions should be improved.

Author Response

1、The paper needs a content reorganization. For example, point 2.2. is more similar to a methodological section rather that a literature review one.

Thank you for this comment. We have moved the section 2.2 to methodological section 3.2. Qualitative interviews.

2、As far as the section 2.3 is concerned, I think that much of the information is more proper of the theoretical section. Thus, I find some confusion in the content presentation.

Thank you for this comment. There is no section 2.3 in the original paper. I'm confused about this comment.

3、I also think that section 4 is divides into too many sections, and that the discourse should be more fluid.

Thank you for this comment. We combined the section of data collection and analysis approach into one part (Data collection and analysis approach, please see lines 238-266). We reorganized the structure of the full paper to make the discourse more fluid, such as merging the section 2(Theoretical background and research model) into other sections of the paper (Introduction and Study Methods).

4、I would advise authors to review the content and to reinforce the theoretical review.

Thank you for this comment. We have read the related literature of SOR once again, and quoted them in the paper, which deepened our understanding of SOR theory (please see lines 364-365).

5、Conclusions should be improved.

Thank you for this comment. Conclusions have been improved (please see lines 436-455).

Reviewer 2 Report

Dear authors

I reviewed the manuscript entitled “What drives visitors’ use of bins in urban parks? An application of the stimulus-organism-response paradigm”. This study is of important topic. In my opinion, it is fall within the scope of journal. It should also be mentioned that the manuscript has addressed an issue that is my favorite subject area. Reading and evaluating different parts of the paper shows that the authors have made a lot of effort to carry out this research. Their efforts have resulted in important and ground-breaking conclusions that can certainly be used by different end-users. Therefore, I recommend this paper for publication. However, there are some points that should be addressed by the respected authors before consideration of the manuscript for publication in IJERPH. My main comments are as follows:

1.       In the title of the manuscript the authors have used the term “paradigm”. However, “the stimulus-organism-response” is a “model”. Please replace it with the term “model”.

2.       It would be great if the number of the statistical population of the study and sampling method be mentioned in the abstract section.

3.       Please highlight one of the most important recommendations of the study in the end of abstract.

4.       In the introduction section, respected authors have mentioned that “We used the theory of stimulus-organism-response (SOR; Mehrabianand Russell, 1974) to explain visitors’ use of bins.” However, logical justification should be presented for employing this model. Since, as the authors have mentioned, there so many competing theories that could be used. However, the have employed stimulus-organism-response model. Why? What were the main superiorities of this model?

5.       In the introduction section, the respected authors have mentioned that:

“We used a mixed methods approach to identify related stimuli, facilitators, and inhibitors that may drive visitors’ use of bins. First, we conducted a qualitative study through in-depth interviews to find out why park visitors felt that they should or should not put their rubbish in a bin. We collected data from 356 visitors across three urban parks in Shanghai to test our hypotheses.”

“We analyzed the results of the interviews and, in combination with a comprehensive literature review, determined that personal and social norms affected the use of bins. We determined that these could either promote or inhibit people’s use of bins. In terms of our SOR framework, the stimuli were personal and social norms; the internal states of organisms were the facilitators and inhibitors of the use of bins; and the use of bins was the potential response.”

These two paragraphs are related to the methodology and should be moved to the methods section.

6.       In the last two paragraphs of the manuscript, the authors have explained the research questions and also the originalities. It is great and well done. However, I believe that the relationships and the path mechanisms hypothesized should be visualized using a conceptual framework in the end of the introduction section.

7.       This study has been done in China. However, I recommend the respected authors to highlight the global value of this research in the end of introduction section. Can the results of this study be applied to encourage the usage of bins in other parts of the world?

8.       In the last paragraph you have mentioned the originalities. However, it would be great if you try to categorize your originalities in terms of contributions to “theory” and “practice”.

9.       Since you have presented a conceptual framework in the end of part 2, please ignore the comment 6 of mine.

10.   I have recently read some papers that have used personal norms as one of the predictors of intention and behavior. You can cite them in part 2 or discussion section. Please see the following paper:

Valizadeh, N., Bijani, M., & Abbasi, E. (2021). Farmers’ participatory-based water conservation behaviors: evidence from Iran. Environment, Development and Sustainability, 23(3), 4412-4432.

Lee, S., Park, H. J., Kim, K. H., & Lee, C. K. (2021). A moderator of destination social responsibility for tourists’ pro-environmental behaviors in the VIP model. Journal of Destination Marketing & Management20, 100610.

de Groot, J. I., Bondy, K., & Schuitema, G. (2021). Listen to others or yourself? The role of personal norms on the effectiveness of social norm interventions to change pro-environmental behavior. Journal of Environmental Psychology78, 101688.

Valizadeh, N., Bijani, M., & Abbasi, E. (2018). Farmers active participation in water conservation: insights from a survey among farmers in southern regions of West Azerbaijan Province, Iran. Journal of Agricultural Science and Technology, 20(5), 895-910.

11.   Results has been written and articulated very well. There is no need for further revisions.

12.   Discussions and conclusions section has been written and articulated very well and the methods completely support the results. However, in discussion section please try to put your results in an international scope and then provide the readers with some useful global level recommendations. Also, in this section the respected authors should try compare their results with the results of other researchers in China and other parts of the world.

13.   In conclusion section, I recommend the respected authors to mention the main take-home message of the research in a short paragraph.

14.   Please highlight the main limitation of your study and try to draw some future pathways for the future researchers.

15.   In conclusion section try to highlight the main contribution of your paper to the theory and practice.

In general, I believe that this manuscript can be accepted for publication in IJERPH after major revisions. 

Author Response

1.In the title of the manuscript the authors have used the term “paradigm”. However, “the stimulus-organism-response” is a “model”. Please replace it with the term “model”.

Thank you for this comment. We have replaced “paradigm” with the term “model” (please see line 3).

2.  It would be great if the number of the statistical population of the study and sampling method be mentioned in the abstract section.

Thank you for this comment. We have mentioned the statistical population and sampling method  in abstract section (please see lines 14-15). 

3.  Please highlight one of the most important recommendations of the study in the end of abstract.

Thank you for this comment. We have highlighted one of the most important recommendations of the study in the end of abstract (please see line 21-24). 

4.In the introduction section, respected authors have mentioned that “We used the theory of stimulus-organism-response (SOR; Mehrabianand Russell, 1974) to explain visitors’ use of bins.” However, logical justification should be presented for employing this model. Since, as the authors have mentioned, there so many competing theories that could be used. However, the have employed stimulus-organism-response model. Why? What were the main superiorities of this model?

Thank you for this comment. We have elaborated the main superiorities of this model in the introduction (please see lines 78-79).

5.In the introduction section, the respected authors have mentioned that:“We used a mixed methods approach to identify related stimuli, facilitators, and inhibitors that may drive visitors’ use of bins. First, we conducted a qualitative study through in-depth interviews to find out why park visitors felt that they should or should not put their rubbish in a bin. We collected data from 356 visitors across three urban parks in Shanghai to test our hypotheses.” “We analyzed the results of the interviews and, in combination with a comprehensive literature review, determined that personal and social norms affected the use of bins. We determined that these could either promote or inhibit people’s use of bins. In terms of our SOR framework, the stimuli were personal and social norms; the internal states of organisms were the facilitators and inhibitors of the use of bins; and the use of bins was the potential response.” These two paragraphs are related to the methodology and should be moved to the methods section.

Thank you for this comment. These two paragraphs were moved and integrated into the methods section.

6. In the last two paragraphs of the manuscript, the authors have explained the research questions and also the originalities. It is great and well done. However, I believe that the relationships and the path mechanisms hypothesized should be visualized using a conceptual framework in the end of the introduction section.

Thank you for this comment. The path mechanisms hypothesized have been visualized using a conceptual framework in the end of the introduction section (please see Figure.1).

7.This study has been done in China. However, I recommend the respected authors to highlight the global value of this research in the end of introduction section. Can the results of this study be applied to encourage the usage of bins in other parts of the world?

Thank you for this comment. We have highlighted the global value of this research in the introduction section (please see lines 97-99).

8.In the last paragraph you have mentioned the originalities. However, it would be great if you try to categorize your originalities in terms of contributions to “theory” and “practice”.

Thank you for this comment. We have categorized originalities in terms of contributions to “theory” and “practice” in the section of conclusion (please see lines 444-454).

9. Since you have presented a conceptual framework in the end of part 2, please ignore the comment 6 of mine.

Thank you for this comment.

10.I have recently read some papers that have used personal norms as one of the predictors of intention and behavior. You can cite them in part 2 or discussion section. Please see the following paper:Valizadeh, N., Bijani, M., & Abbasi, E. (2021). Farmers’ participatory-based water conservation behaviors: evidence from Iran. Environment, Development and Sustainability, 23(3), 4412-4432.Lee, S., Park, H. J., Kim, K. H., & Lee, C. K. (2021). A moderator of destination social responsibility for tourists’ pro-environmental behaviors in the VIP model. Journal of Destination Marketing & Management20, 100610. de Groot, J. I., Bondy, K., & Schuitema, G. (2021). Listen to others or yourself? The role of personal norms on the effectiveness of social norm interventions to change pro-environmental behavior. Journal of Environmental Psychology78, 101688.Valizadeh, N., Bijani, M., & Abbasi, E. (2018). Farmers active participation in water conservation: insights from a survey among farmers in southern regions of West Azerbaijan Province, Iran. Journal of Agricultural Science and Technology, 20(5), 895-910.

Thank you for this comment. We have cited the paper in the introduction and discussion (please see lines 81, 364-365). 

11.Results has been written and articulated very well. There is no need for further revisions.

Thank you for this comment.

12.Discussions and conclusions section has been written and articulated very well and the methods completely support the results. However, in discussion section please try to put your results in an international scope and then provide the readers with some useful global level recommendations. Also, in this section the respected authors should try compare their results with the results of other researchers in China and other parts of the world.

Thank you for this comment. (1)We have put your results in an international scope and then provide the readers with some useful global level in the discussions and conclusions section (please see lines 363-367).

(2) We have compared the results with the results of other researchers in China and other parts of the world (please see lines 340,355-356,381-382).

13.   In conclusion section, I recommend the respected authors to mention the main take-home message of the research in a short paragraph.

Thank you for this comment. We mentioned the main take-home message of the research in the conclusion section (please see lines 440-454).

14.Please highlight the main limitation of your study and try to draw some future pathways for the future researchers.

Thank you for this comment. We have highlighted the main limitation of the study and draw some future pathways for the future researchers (please see lines 424-435).

15.In conclusion section try to highlight the main contribution of your paper to the theory and practice.

Thank you for this comment. We have highlighted the main contribution of the paper to the theory and practice (please see lines 444-454).

Reviewer 3 Report

The manuscript titled “What drives visitors’ use of bins in urban parks? An application of the stimulus-organism-response paradigm” aimed to explore the mechanism that drives people’s use of bins in urban parks, applying the theoretical model of stimulus-organism-response.

I carefully read the manuscript and I suppose it may be of high interest for readers of IJERPH. Even so, before publishing it, it could be worth considering some points. Below there are some comments and suggestions.

Line 67: It would be very useful to further explain the mentioned theory, including some examples of previous applications and why authors decided to apply it to explain visitors’ use of bins. I would move paragraph 2.1. here and integrate it.

Line 107: Please report some information on the sample of visitors involved in the qualitative interviews.

Line 253: Performing a power analysis may be appropriate.

It is not perfectly clear to me what are the facilitators and what are the inhibitors. Please make it clearer.

Mediation model: Are the variable latent or not? How did authors treat this variable? Does the mediation model include a CFA for each latent variable? If not, please clarify these statistical aspects.

In general, I strongly suggest splitting the present work in two studies, one based on qualitative interviews and psychometric properties of the questionnaire, and one based on testing the mediation model. This would be beneficial for the readability of the entire work.

Author Response

1.Line 67: It would be very useful to further explain the mentioned theory, including some examples of previous applications and why authors decided to apply it to explain visitors’ use of bins. I would move paragraph 2.1. here and integrate it.

Thank you for this comment.We have moved paragraph 2.1 to the introduction section and integrate it (please see lines 72-81).

2. Line 107: Please report some information on the sample of visitors involved in the qualitative interviews.

Thank you for this comment.We have reported some information on the sample of visitors involved in the qualitative interviews (please see lines 189-192).

3.Line 253: Performing a power analysis may be appropriate.

Thank you for this comment. We have used the other method to prove sufficient sample size (please see lines 255-258). 

4.It is not perfectly clear to me what are the facilitators and what are the inhibitors. Please make it clearer.

Thank you for this comment. We explained the items of facilitators and inhibitors in section 3.3(please see lines 233-234) and table 2.

5.Mediation model: Are the variable latent or not? How did authors treat this variable? Does the mediation model include a CFA for each latent variable? If not, please clarify these statistical aspects

Thank you for this comment.Facilitators and inhibitors are latent variable. In the questionnaire, three observed variables are used to measure these two latent variables respectively. Table 2 shows the all latent variables and the corresponding observed variables.The model include confirmatory factor analysis (CFA) for all the latent variables. Table 2 and “Measurement model” section show the results (please see lines 283-295). The result indicated that the hypothesised model returned a good model fit (χ2/df = 1.49, CFI = 0.99, NFI= 0.97, IFI= 0.99, RFI= 0.97,TLI=0.99, RMSEA= 0.04) (please see lines 282-294).

6.In general, I strongly suggest splitting the present work in two studies, one based on qualitative interviews and psychometric properties of the questionnaire, and one based on testing the mediation model. This would be beneficial for the readability of the entire work.

Thank you for this comment. Qualitative interview and psychometric properties of the questionnaire are closely combined. The hypothetical model is obtained through qualitative analysis, and the model is measured through questionnaire survey, which is a complete research. The mediation model is an important part of the whole research model, so it is more appropriate to take the mediation model as a part of the whole model.

Round 2

Reviewer 2 Report

Dear authors

Thank you very much for efforts to address my comments. I believe that there is nor need for further revisions and you r manuscript can be accepted for publication. 

Bests,

Reviewer

Reviewer 3 Report

I am satisfied with Authors' replies to my report, thus, I endorse the publication in the present version.